# Q-ViT: Accurate and Fully Quantized Low-bit Vision Transformer

**Yanjing Li[1][†], Sheng Xu[1][†], Baochang Zhang[1,2], Xianbin Cao[1][*], Peng Gao[3], Guodong Guo[4,5]**

[1]Beihang University, Beijing, P.R.China
[2]Zhongguancun Laboratory, Beijing, P.R.China
[3]Shanghai Artificial Intelligence Laboratory, Shanghai, P.R.China
[4] Institute of Deep Learning, Baidu Research, Beijing, P.R.China
[5] National Engineering Laboratory for Deep Learning Technology and Application,
Beijing, P.R.China
{yanjingli, shengxu, bczhang, xbcao}@buaa.edu.cn

## Abstract

The large pre-trained vision transformers (ViTs) have demonstrated remarkable performance on various visual tasks, but suffer from expensive computational and memory cost problems when deployed on resource-constrained devices. Among the powerful compression approaches, quantization extremely reduces the computation and memory consumption by low-bit parameters and bit-wise operations. However, low-bit ViTs remain largely unexplored and usually suffer from a significant performance drop compared with the real-valued counterparts. In this work, through extensive empirical analysis, we first identify the bottleneck for severe performance drop comes from the information distortion of the low-bit quantized self-attention map. We then develop an information rectification module (IRM) and a distribution guided distillation (DGD) scheme for fully quantized vision transformers (Q-ViT) to effectively eliminate such distortion, leading to a fully quantized ViTs. We evaluate our methods on popular DeiT and Swin backbones. Extensive experimental results show that our method achieves a much better performance than the prior arts. For example, our Q-ViT can theoretically accelerates the ViT-S by $6.14\times$ and achieves about 80.9% Top-1 accuracy, even surpassing the full-precision counterpart by 1.0% on ImageNet dataset. Our codes and models are attached on `https://github.com/YanjingLi0202/Q-ViT`.

## 1   Introduction

Inspired by the success in natural language processing (NLP), transformer-based models have shown great power in various computer vision (CV) tasks, such as image classification [4] and object detection [2]. Pre-trained with large-scale data, these models usually have a tremendous number of parameters. For example, there are 632M parameters taking up 2528MB memory usage and 162G FLOPs in the ViT-H model, which is both memory and computation expensive during inference. This limits these models for the deployment on resource-limited platforms. Therefore, compressed transformers are urgently needed for real applications.

Substantial efforts have been made to compress and accelerate neural networks for efficient online inference. Methods include compact network design [10], network pruning [9], low-rank decomposition [3], quantization [21, 30, 32], and knowledge distillation [24, 31]. Quantization is particularly suitable for deployment on AI chips because it reduces the bit-width of network parameters and activations for efficient inference. Prior post-training quantization (PTQ) methods [18, 14] on ViTs

---

† Equal contribution. ∗ Corresponding author.

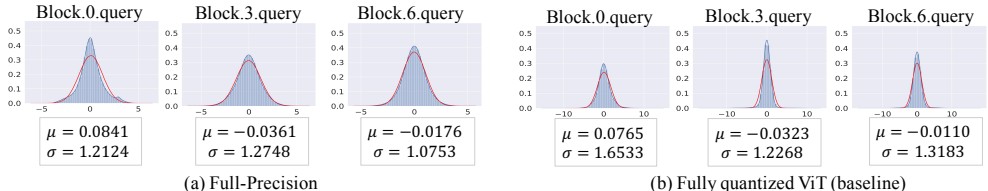

Figure 1: The histogram of query values **q** (blue shadow) along with the PDF curve (red line) of Gaussian distribution $N(\mu, \sigma^2)$ [20], for 3 selected layers in DeiT-T and 4-bit fully quantized DeiT-T (baseline). $\mu$ and $\sigma^2$ are the statistical mean and variance of the values.

directly compute quantized parameters based on pre-trained full-precision models, which constrains the model performance to a sub-optimized level without fine-tuning. Furthermore, quantizing these models based on PTQ methods to ultra-low bits (*e.g.*, 4 bits or lower) is ineffective and suffers from a significant performance reduction.

Differently, quantization-aware training (QAT) [16] methods perform quantization during back propagation and achieve much less performance drop with a higher compression rate generally. QAT is shown to be effective for CNN models [17] for CV tasks. However, QAT methods remain largely unexplored for low-bit quantization of vision transformers. Therefore, we first build a fully quantized ViT baseline, a straightforward yet effective solution based on common techniques. Our study discovers that the performance drop of fully quantized ViT lies in the information distortion among the attention mechanism in the forward process, and the ineffective optimization for eliminating the distribution difference through distillation in the backward propagation. First, the attention mechanism of ViT aims at modeling long-distance dependencies [27, 4]. However, our analysis shows that a direct quantization method leads to the information distortion, *i.e.*, significant distribution variation for the query module between quantized ViT and full-precision counterpart. For example, as shown in Fig. 1, the variance difference is 0.4409 (1.2124 *v.s.* 1.6533) for the first block . This inevitably deteriorates the representation capability of the attention module on capturing the global dependency for the input. Second, the distillation for the fully quantized ViT baseline utilizes distillation token (following [25]) to directly supervise the classification output of the quantized ViT. However, we found that such a simple supervision is ineffective, which is coarse-grained for the large gap between the quantized attention scores and their full-precision counterparts.

To address the aforementioned issues, a fully quantized ViT (Q-ViT) is developed by retaining the distribution of quantized attention modules as that of full-precision counterparts (see the overview in Fig. 2). Accordingly, we propose to modify the distorted distribution over quantized attention modules through an Information Rectification Module (IRM) based on information entropy maximization, in the forward process. While in the backward process, we present a Distribution Guided Distillation (DGD) scheme to eliminate the distribution variation through attention similarity loss between quantized ViT and full-precision counterpart. The contributions of our work include:

- We propose an Information Rectification Module (IRM) based on the information theory to address the information distortion problem. IRM applies quantized representations in the attention module with a maximized information entropy, allowing the quantized model to restore the representation of input images.

- We develope a Distribution Guided Distillation (DGD) scheme to eliminate the distribution mismatch in distillation. DGD takes appropriate activations and utilizes knowledge from the similarity matrices in distillation to perform optimization accurately.

- Our Q-ViT, for the first time, explores a promising way towards accurate and low-bit ViT. Extensive experiments on the ImageNet benchmark show that our Q-ViT outperforms the baseline by a large margin, and achieves comparable performances with the full-precision counterparts.

---

The Gaussian distribution hypothesis is supported by [20]

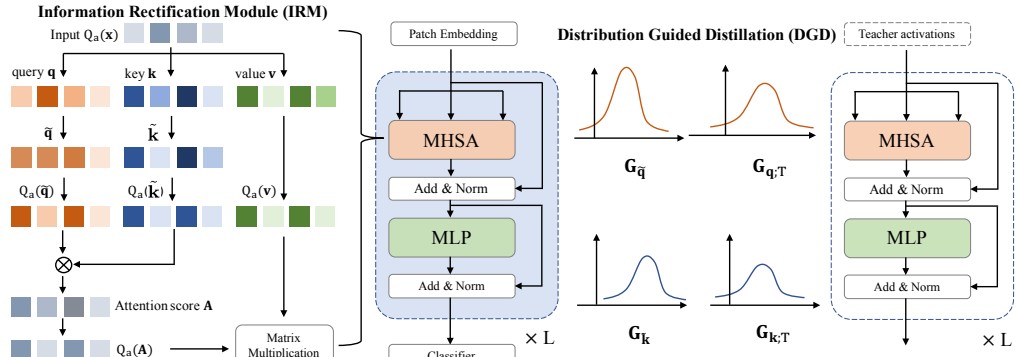

Figure 2: Overview of Q-ViT, applying Information Rectification Module (IRM) for maximizing representation information and Distribution Guided Distillation (DGD) for accurate optimization.

## 2   Related Work

**Vision transformer.** Motivated by the great success of Transformer in natural language processing, researchers are trying to apply Transformer architecture to Computer Vision tasks. Unlike mainstream CNN-based models, Transformer is capable of capturing long-distance visual relations by its self-attention module and provides the paradigm without image-specific inductive bias. ViT [4] views $16 \times 16$ image patches as token sequence and predicts classification via a unique class token, which shows promising results. Subsequently, many works, such as DeiT [25] and PVT [28] achieve further improvement on ViT, making it more efficient and applicable in downstream tasks. CONTAINER [7] fully utilizes a hybrid ViT to aggregate dynamic and static information, exploring a new framework for visual tasks. However, these high-performing vision transformers are attributed to the large number of parameters and high computational overhead, limiting their adoption. Therefore, innovating a smaller and faster vision transformer becomes a new trend. DynamicViT [23] presents a dynamic token sparsification framework to prune redundant tokens progressively and dynamically, achieving competitive complexity and accuracy trade-off. Evo-ViT [33] proposes a slow-fast updating mechanism that guarantees information flow and spatial structure, trimming down both the training and inference complexity. While the above works focus on efficient model designing, this paper boosts the compression and acceleration in the track of quantization.

**Quantization.** Quantizing neural networks (QNNs) often possess low-bit ($1 \sim$ 4-bit) weights and activations to accelerate the model inference and save the memory usage. Specifically, ternary weights are introduced to reduce the quantization error in TWN [13]. DoReFa-Net [35] exploits convolution kernels with low bit-width parameters and gradients to accelerate both the training and inference. TTQ [36] uses two full-precision scaling coefficients to quantize the weights to ternary values. [37] presented a $2 \sim$ 4-bit quantization scheme using a two-stage approach to alternately quantize the weights and activations, which provides an optimal trade-off among memory, efficiency, and performance. [11] parameterizes the quantization intervals and obtain their optimal values by directly minimizing the task loss of the network and also the accuracy degeneration with further bit-width reduction. [29] introduces transfer learning into network quantization to obtain an accurate low-precision model by utilizing the Kullback-Leibler (KL) divergence. [6] enables accurate approximation for tensor values that have bell-shaped distributions with long tails and finds the entire range by minimizing the quantization error. In our Q-ViT, we aim to implement an accurate, fully quantized vision transformer under the QAT paradigm.

## 3   Baseline of Fully Quantized ViT

First of all, we build a baseline to study the fully quantized ViT since it has never been proposed in previous works. A straightforward solution is to quantize the representations (weights and activations) in ViT architecture in the forward propagation and apply distillation to the optimization in the backward propagation.

**Quantized ViT architecture.** We briefly introduce the technology of neural network quantization. We first introduce a general asymmetric activation quantization and symmetric weight quantization

scheme as

$$Q_a(x) = \lfloor \text{clip}\{(x-z)/\alpha_x, -Q_n^x, Q_p^x\} \rceil \quad Q_w(\mathbf{w}) = \lfloor \text{clip}\{\mathbf{w}/\alpha_\mathbf{w}, -Q_n^\mathbf{w}, Q_p^\mathbf{w}\} \rceil$$
$$\hat{x} = Q_a(x) \times \alpha_x + z, \qquad\qquad\qquad \hat{\mathbf{w}} = Q_w(\mathbf{w}) \times \alpha_\mathbf{w}. \tag{1}$$

Here, $\text{clip}\{y, r_1, r_2\}$ returns $y$ with values below $r_1$ set as $r_1$ and values above $r_2$ set as $r_2$, and $\lfloor y \rceil$ rounds $y$ to the nearest integer. With quantizing activations to signed $a$ bits and weights to signed $b$ bits, $Q_n^x = 2^{a-1}, Q_p^x = 2^{a-1} - 1$ and $Q_n^\mathbf{w} = 2^{b-1}, Q_p^\mathbf{w} = 2^{b-1} - 1$. In general, the forward and backward propagation of quantization function in quantized network is formulated as

$$\text{Forward:} \quad \text{Q-Linear}(x) = \hat{x} \cdot \hat{\mathbf{w}} = \alpha_x \alpha_\mathbf{w}((Q_a(x) + z/\alpha_x) \otimes Q_w(\mathbf{w})),$$

$$\text{Backward:} \quad \frac{\partial \mathcal{J}}{\partial x} = \frac{\partial \mathcal{J}}{\partial \hat{x}} \frac{\partial \hat{x}}{\partial x} = \begin{cases} \frac{\partial \mathcal{J}}{\partial \hat{x}} & \text{if } x \in [-Q_n^x, Q_p^x] \\ 0 & \text{otherwise} \end{cases},$$

$$\frac{\partial \mathcal{J}}{\partial \mathbf{w}} = \frac{\partial \mathcal{J}}{\partial x} \frac{\partial x}{\partial \hat{\mathbf{w}}} \frac{\partial \hat{\mathbf{w}}}{\partial \mathbf{w}} = \begin{cases} \frac{\partial \mathcal{J}}{\partial x} \frac{\partial x}{\partial \hat{\mathbf{w}}} & \text{if } \mathbf{w} \in [-Q_n^\mathbf{w}, Q_p^\mathbf{w}] \\ 0 & \text{otherwise} \end{cases}, \tag{2}$$

where $\mathcal{J}$ is loss function, $Q(\cdot)$ is applied in the forward propagation while the straight-through estimator (STE) [1] is used to retain the derivation of gradient in backward propagation. $\otimes$ denotes the matrix multiplication with efficient bit-wise operations.

The input images are first encoded as patches and passes through several transformer blocks. Such transformer block consists of two components: Multi-Head Self-Attention (MHSA) and Multi-Layer Perceptron (MLP). The computation of attention weight depends on the corresponding query $\mathbf{q}$, key $\mathbf{k}$ and value $\mathbf{v}$, and the quantized computation in one attention head is

$$\mathbf{q} = \text{Q-Linear}_q(x), \mathbf{k} = \text{Q-Linear}_k(x), \mathbf{v} = \text{Q-Linear}_v(x), \tag{3}$$

where $\text{Q-Linear}_q$, $\text{Q-Linear}_k$, $\text{Q-Linear}_v$ denote the three quantized linear layers for $\mathbf{q}, \mathbf{k}, \mathbf{v}$, respectively. Thus, the attention weight is formulated as

$$\mathbf{A} = \frac{1}{\sqrt{d}}(Q_a(\mathbf{q}) \otimes Q_a(\mathbf{k})^\top),$$
$$Q_\mathbf{A} = Q_a(\text{softmax}(\mathbf{A})). \tag{4}$$

**Training for Quantized ViT.** Knowledge distillation is an essential supervision approach for training quantized neural networks, which bridges the performance gap between quantized models and full-precision counterparts. The usual practice is using distillation through attention as described in [25]

$$\mathcal{L}_\text{dist} = \frac{1}{2}\mathcal{L}_\text{CE}(\psi(Z_q), y) + \frac{1}{2}\mathcal{L}_\text{CE}(\psi(Z_q), y_t),$$
$$y_t = \arg\max_c Z_t(c). \tag{5}$$

## 4 Proposed Q-ViT

With aforementioned quantizing and training pipeline, a fully quantized ViT baseline is built, which however has a large performance gap with full-precision counterparts. Our study shows that the baseline suffers an severe information distortion among quantized attention scores in the forward propagation and distillation direction misleading in the backward propagation.

### 4.1 Performance degeneration of Fully Quantized ViT Baseline

Intuitively, in the fully quantized ViT baseline, the information representation capability largely depends on the transformer-based architecture,

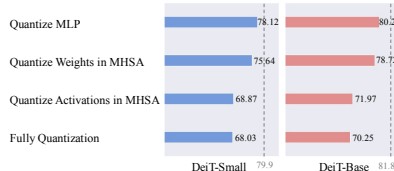

Figure 3: Analysis of bottleneck from architecture perspective. We report the accuracy of 2-bit quantized DeiT-S and DeiT-B on ImageNet dataset about replacing full-precision structure.

such as the attention weight in MHSA module. However, the performance improvement brought by such architecture is severely limited by the quantized parameters, while the rounded and discrete quantization also significantly affect the optimization. The phenomenon identifies the bottleneck of the fully quantized ViT baseline comes from architecture and optimization for the forward and backward propagation, respectively.

**Architecture bottleneck.** We replace each module with the full-precision counterpart respectively and compare the accuracy drop as shown in Fig. 3. These ablation experiments are conducted the same as experiments in Tab. 2. We find that quantizing query, key, value and attention weight (*i.e.*, softmax($\mathbf{A}$) in Eq. (4) to 2 bits brings the most significant drop of accuracy among all parts of the ViT, up to 10.03%. While quantized MLP layers and quantized weights of linear layers in MHSA brings only 1.78% and 4.26% drop, respectively. And once query, key, value and attention weight are quantized, even keep all weights of linear layers in MHSA module full-precision, the performance drops (10.57%) are still significant. Thus, improving the attention structure is pivotal to solve the performance drop problem of quantized ViT.

**Optimization bottleneck.** We calculate l2-norm distances between each attention weight among different blocks in DeiT-S architecture as shown in Fig. 4. The MHSA modules in full-precision ViT with different depth learn different representations from images. As mentioned in [22], lower ViT layers attend both locally and globally while higher ViT layers pay most attention to global representations. However, the fully quantized ViT (blue lines in Fig. 4) fails to learn accurate attention map distances. Thus, it requires a new design that could utilize the full-precision teacher's information better.

### 4.2 Information Rectification in Q-Attention

To address the information distortion of quantized representations in the forward propagation, we propose an efficient Q-Attention structure based on information theory, which statistically maximizes the entropy of representation and revives the attention mechanism in the fully quantized ViT. Since the representations with extremely compressed bit-width in fully quantized ViT have limited capabilities, the ideal quantized representation should preserve the given full-precision counterparts as much as possible, which means the mutual information between quantized and full-precision representations should be maximized as mentioned in [20].

We further show the statistical results that the distribution of query and key values in ViT architectures intending to follow Gaussian distributions under the distilling supervision, whose histograms are in bell-shape [20]. For example, in Fig. 1 and Fig. 5, we have shown the query and key distributions and their corresponding Probability Density Function (PDF) using the calculated mean and standard deviation for each MHSA layer. Therefore, the distributions of query and key in the MHSA modules of full-precision counterparts are formulated as

$$\mathbf{q} \sim \mathcal{N}(\mu(\mathbf{q}), \sigma(\mathbf{q})), \quad \mathbf{k} \sim \mathcal{N}(\mu(\mathbf{k}), \sigma(\mathbf{k})). \tag{6}$$

Since the weight and the activation with extremely compressed bit-width in fully quantized ViT have limited capabilities, the ideal quantization process should preserve the corresponding full-precision counterparts as much as possible, thus the mutual information between quantized and full-precision representations should be maximized [20]. As shown in [19], for Gaussian distribution, the quantizers with maximum output entropy (MOE) and minimum average error (MAE) are approximately the same within a multiplicative constant. Thus the process of minimizing the error between full-precision values and quantized values is equivalent to maximizing the information entropy of the quantized values. Thus, when the deterministic quantization function is applied to quantized ViT, such objective is equivalent to maximizing the information entropy $\mathcal{H}(Q_{\mathbf{x}})$ of quantized representation $Q_{\mathbf{x}}$ [19] in Eq. (4), which is defined as

$$\mathcal{H}(Q_a(\mathbf{x})) = -\sum_{q_{\mathbf{x}}} p(q_{\mathbf{x}}) \log p(q_{\mathbf{x}}) = \frac{1}{2} \log 2\pi e \sigma_{\mathbf{x}}^2,$$

$$\max \mathcal{H}(Q_a(\mathbf{x})) = \frac{n \ln 2}{2^n}, \quad \text{when} \ \ p(q_{\mathbf{x}}) = \frac{1}{2^n}, \tag{7}$$

where $q_{\mathbf{x}}$ is the random quantized variables in $Q_a(\mathbf{x})$ (which is $Q_a(\mathbf{q})$ or $Q_a(\mathbf{k})$ in different conditions) with probability mass function $p(\cdot)$. For better retaining the information contained in the

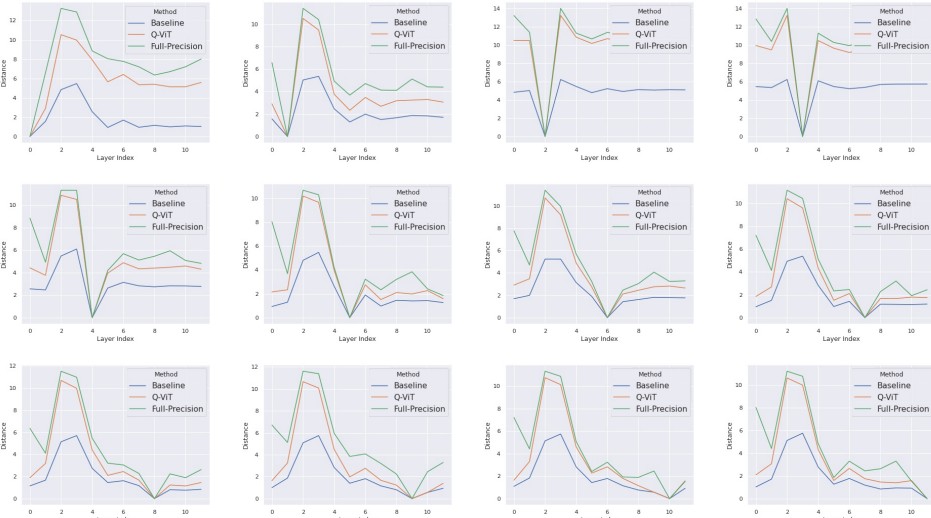

Figure 4: Attention-distance comparison for full-precision DeiT-Small (green lines) , fully quantized DeiT-Small baseline (blue lines), and Q-ViT (yellow lines) for same input. Q-ViT shows similar behavior with the full- precision model, while baseline suffers indistinguishable attention distance for information degradation.

MHSA modules from the full-precision counterparts, the information entropy in the quantization process should be maximized.

However, direct application of quantization function converting the values into finite fixed points brings irreversible disturbance to the distributions and the information entropy $\mathcal{H}(Q_a(\mathbf{q}))$ and $\mathcal{H}(Q_a(\mathbf{k}))$ degenerates to a much lower level than the full-precision counterparts. To mitigate the information degradation from the quantization process in the attention mechanism, a Information Rectification Module (IRM) is proposed for effectively maximizing the information entropy of quantized attention weights

$$Q_a(\tilde{\mathbf{q}}) = Q_a\left(\frac{\mathbf{q} - \mu(\mathbf{q}) + \beta_{\mathbf{q}}}{\gamma_{\mathbf{q}}\sqrt{\sigma^2(\mathbf{q}) + \epsilon_{\mathbf{q}}}}\right), \quad Q_a(\tilde{\mathbf{k}}) = Q_a\left(\frac{\mathbf{k} - \mu(\mathbf{k}) + \beta_{\mathbf{k}}}{\gamma_{\mathbf{k}}\sqrt{\sigma^2(\mathbf{k}) + \epsilon_{\mathbf{k}}}}\right), \tag{8}$$

where $\gamma_{\mathbf{q}}, \beta_{\mathbf{q}}$ and $\gamma_{\mathbf{k}}, \beta_{\mathbf{k}}$ are learnble parameters for modifying the distribution of $\tilde{\mathbf{q}}$, while $\epsilon_{\mathbf{q}}$ and $\epsilon_{\mathbf{k}}$ are constants preventing the denominator being 0. The learning rates of learnable $\gamma_{\mathbf{q}}, \beta_{\mathbf{q}}$ and $\gamma_{\mathbf{k}}, \beta_{\mathbf{k}}$ are same as the whole network. Thus after IRM, the information entropy $\mathcal{H}(Q_a(\tilde{\mathbf{q}}))$ and $\mathcal{H}(Q_a(\tilde{\mathbf{k}}))$ are formulated as

$$\mathcal{H}(Q(\tilde{\mathbf{q}})) = \frac{1}{2}\log 2\pi e[\gamma_{\mathbf{q}}^2(\sigma_{\mathbf{q}}^2 + \epsilon_{\mathbf{q}})], \quad \mathcal{H}(Q(\tilde{\mathbf{k}})) = \frac{1}{2}\log 2\pi e[\gamma_{\mathbf{k}}^2(\sigma_{\mathbf{k}}^2 + \epsilon_{\mathbf{k}})]. \tag{9}$$

Then to revive the attention mechanism to capture critic elements by information entropy maximization, the learnable parameters $\gamma_{\mathbf{q}}, \beta_{\mathbf{q}}$ and $\gamma_{\mathbf{k}}, \beta_{\mathbf{k}}$ reshape the distributions of the query and key values to achieve the state of information maximization. In a nutshell, in our IRM-Attention structure, the information entropy of quantized attention weight is maximized to alleviate its severe information distortion and revive the attention mechanism.

## 4.3 Distribution Guided Distillation through Attention

To address the attention distribution mismatch occurred in fully quantized ViT baseline in the backward propagation, we further propose a Distribution Guided Distillation (DGD) scheme with apposite distilled activations and the well-designed similarity matrices to effectively utilize knowledge from the teacher, which optimizes the fully quantized ViT more accurately.

As an optimization technique based on element-level comparison of activation, distillation allows the quantized ViT to mimic the full-precision teacher model about output logits. However, we find that the

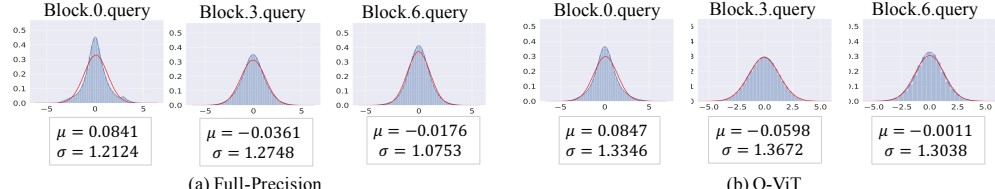

(a) Full-Precision            (b) Q-ViT

Figure 5: The histogram of query and key values $\mathbf{q}, \mathbf{k}$ (blue shadow) along with the PDF curve (red line) of Gaussian distribution $N(\mu, \sigma^2)$ [20], for 3 selected layers in DeiT-T and 4-bit Q-ViT. $\mu$ and $\sigma^2$ are the statistical mean and variance of the values.

distillation procedure used in previous ViT and fully quantized ViT baseline (Sec. 3) unable to deliver meticulous supervision to attention weights (shown in Fig. 4), leading to insufficient optimization. To solve the optimization insufficiency in the distillation of the fully quantized ViT, we propose the Distribution-Guided Distillation (DGD) method in Q-ViT. We first build patch-based similarity pattern matrices for distilling the upstream query and key instead of attention following [26], which is formulated as

$$\tilde{G}^l_{\mathbf{q}_h} = \tilde{\mathbf{q}}^l_h \cdot (\tilde{\mathbf{q}}^l_h)^\top, \;\; G^{(l)}_{\mathbf{q}_h} = \tilde{G}^l_{\mathbf{q}_h}/\|\tilde{G}^l_{\mathbf{q}_h}\|_2,$$
$$\tilde{G}^l_{\mathbf{k}_h} = \tilde{\mathbf{k}}^l_h \cdot (\tilde{\mathbf{k}}^l_h)^\top, \;\; G^{(l)}_{\mathbf{k}_h} = \tilde{G}^l_{\mathbf{k}_h}/\|\tilde{G}^l_{\mathbf{k}_h}\|_2, \tag{10}$$

where $\|\cdot\|_2$ denotes $\ell_2$ normalization, and $l, h$ are layer index and head index. Previous work shows that matrices constructed in this way are regarded as the specific patterns reflecting the semantic comprehension of network [26]. And the patches encoded from input images containing high-level understanding of parts, objects, and scenes [8]. Thus such semantic-level distillation target provide guided and meticulous supervision to the quantized ViT. The corresponding $\tilde{G}^l_{\mathbf{q}_h;T}$ and $\tilde{G}^l_{\mathbf{k}_h;T}$ are constructed in the same way by the teacher's activation. Thus combining the original distillation loss in Eq. (5), the final distillation loss is then formulated as

$$\mathcal{L}_{\text{DGD}} = \sum_{l \in [1,L]} \sum_{h \in [1,H]} \|\tilde{G}^l_{\mathbf{q}_h;T} - \tilde{G}^l_{\mathbf{q}_h}\|_2 + \|\tilde{G}^l_{\mathbf{k}_h;T} - \tilde{G}^l_{\mathbf{k}_h}\|_2,$$
$$\mathcal{L}_{\text{distillation}} = \mathcal{L}_{\text{dist}} + \mathcal{L}_{\text{DGD}}, \tag{11}$$

where $L$ and $H$ denote the number of ViT layers and heads. With the proposed Distribution Guided Distillation, the Q-ViT retains the distribution over query and key from the full-precision counterparts (as shown in Fig. 5).

Our DGD scheme first provides the distribution-aware optimization direction together with processing appropriate distilled parameters and then constructs similarity matrices to eliminate scale differences and numerical instability, thereby improves fully quantized ViT by accurate optimization.

## 5 Experiments

In this section, we evaluate the performance of the proposed Q-ViT model for image classification task using popular DeiT [25] and Swin [15] backbones. To the best of our knowledge, there is no publicly available source code on quantization-aware training of vision transformer at this point, so we implement the baseline and LSQ [5] methods by ourselves.

### 5.1 Datasets and Implementation Details

**Datasets.** The experiments are carried out on the ILSVRC12 ImageNet classification dataset [12]. The ImageNet dataset is more challenging due to its large scale and greater diversity. There are 1000 classes and 1.2 million training images, and 50k validation images in it. In our experiments, we use the classic data augmentation method described in [25].

**Experimental settings.** In our experiments, we initialize the weights of quantized model with the corresponding pretrained full-precision model. The quantized model is trained for 300 epochs with batch-size 512 and the base learning rate $2e-4$. We do not use warm-up scheme. For all the experiments, we apply LAMB [34] optimizer with weight decay set as 0. Other training settings

follow DeiT [25] or Swin Transformer [15]. Note that we use 8-bit for the patch embedding (first) layer and the classification (last) layer following [5].

**Bakcbone.** We evaluate our quantization method on two popular vision transformer implementation: DeiT [25] and Swin Transformer [15]. The DeiT-S, DeiT-B, Swin-T and Swin-S are adopted as the backbone models, whose Top-1 accuracy on ImageNet dataset are 79.9%, 81.8%, 81.2%, and 83.2% respectively. For a fair comparison, we utilize the official implementation of DeiT and Swin Transformer.

## 5.2 Ablation Study

We give quantitative results of the proposed IRM and DGD in Tab. 1 As shown in Tab. 1, the fully quantized ViT baseline suffers a severe performance drop on classification task (0.2%, 2.1% and 11.7% with 2/3/4-bit, respectively). IRM and DGD improve the performance when used alone, and the two techniques further boost the performance considerably when combined together. For example, the IRM improve the 2-bit Baseline by 1.7% and the DGD achieves 2.3% performance improvement. While combining the IRM and DGD together, the performance improvement achieves 3.8%.

Table 1: Evaluating the components of Q-ViT based on ViT-S backbone.

| Method | #Bits | Top-1 | #Bits | Top-1 | #Bits | Top-1 |
|---|---|---|---|---|---|---|
| Full-precision | 32-32 | 79.9 | - | - | - | - |
| Baseline | 4-4 | 79.7 | 3-3 | 77.8 | 2-2 | 68.2 |
| +IRM | 4-4 | 80.2 | 3-3 | 78.2 | 2-2 | 69.9 |
| +DGD | 4-4 | 80.4 | 3-3 | 78.5 | 2-2 | 70.5 |
| **+IRM+DGD (Q-ViT)** | **4-4** | **80.9** | **3-3** | **79.0** | **2-2** | **72.0** |

To conclude, the two techniques can promote each other to improve Q-ViT and close the performance gap between fully quantized ViT and full-precision counterpart.

## 5.3 Main Results

The experimental results are shown in Tab. 2. We compare our method with 2/3/4-bit baseline and LSQ [5] based on the same frameworks for the task of image classification with the ImageNet dataset. We also report the classification performance of the 8-bit post-training quantization networks percentile VT-PTQ [18]. We firstly evaluate the proposed method on DeiT-S and DeiT-B models.

For DeiT-S backbone, compared with 8-bit VT-PTQ method, our 4bit Q-ViT achieves a much larger compression ratio than 8-bit VT-PTQ, but with significant performance improvement (78.1% *vs.* 80.9%). And it is worth noting that the proposed 2-bit model significantly compresses the DeiT-S by $21.5\times$ on FLOPs. The proposed method boosts the performance of 2/3/4-bit Baseline by 3.9%, 1.5% and 1.2% with the same architecture and bit-width, which is significant on the ImageNet dataset. For larger DeiT-B, as shown in Tab. 2, the performance of the proposed method outperforms the 2/3/4-bit Baseline by 3.8%, 1.7% and 1.9%, a large margin. Also note that the proposed 2/3/4-bit model significantly compresses the DeiT-B by $21\times$, $12\times$ and $7.6\times$ on FLOPs. Compared with 8-bit post-training quantization methods, our method achieves significantly higher compression rate, and the performance improvement is significant.

Also, our method generates convincing results on Swin-transformers. As shown in Tab. 2, the performance of the proposed method with Swin-T outperforms the 2/3/4-bit Baseline method by 4.1% , 2.1% and 2.0%, a large margin. Compared with 8-bit post-training quantization methods, our method achieves significantly higher compression rate, and comparable performance. Note that our 4-bit Q-ViT surpasses the full-precision by 1.3% counterpart using Swin-T, which demonstrates the significance of our Q-ViT. For larger Swin-S, the performance of the proposed method outperforms the 2/3/4-bit Baseline by 4.3%, 1.8% and 1.5%. Also note that our 4-bit Q-ViT surpasses the full-precision by 1.1% counterpart using Swin-S and significantly compresses the Swin-S by $7.9\times$ , which demonstrates the effectiveness and efficiency of our Q-ViT.

Table 2: Quantization results on ImageNet dataset. "#Bits" (W-A) is the bit width for weights and activation.

| Network | Method | #Bits | Size$_{(MB)}$ | FLOPs$_{(G)}$ | Top-1 | Top-5 |
|---------|--------|-------|---------------|---------------|-------|-------|
| | Full-precision | 32-32 | 88.2 | 4.3 | 79.9 | 95.0 |
| | VT-PTQ | 8$_{MP}$-8$_{MP}$ | 22.2 | - | 78.1 | - |
| | LSQ | 4-4 | 11.4 | 0.7 | 79.6 | 94.6 |
| | Baseline | 4-4 | 11.4 | 0.7 | 79.7 | 94.5 |
| | **Q-ViT** | 4-4 | 11.4 | 0.7 | **80.9** | **94.9** |
| DeiT-S | LSQ | 3-3 | 8.7 | 0.4 | 77.3 | 93.0 |
| | Baseline | 3-3 | 8.7 | 0.4 | 77.5 | 93.3 |
| | **Q-ViT** | 3-3 | 8.7 | 0.4 | **79.0** | **94.2** |
| | LSQ | 2-2 | 6.0 | 0.2 | 68.0 | 86.4 |
| | Baseline | 2-2 | 6.0 | 0.2 | 68.2 | 86.5 |
| | **Q-ViT** | 2-2 | 6.0 | 0.2 | **72.1** | **90.3** |
| | Full-precision | 32-32 | 346.2 | 16.8 | 81.8 | 95.6 |
| | VT-PTQ | 8$_{MP}$-8$_{MP}$ | 86.8 | - | 81.3 | - |
| | LSQ | 4-4 | 44.1 | 2.2 | 80.9 | 95.1 |
| | Baseline | 4-4 | 44.1 | 2.2 | 81.1 | 95.3 |
| | **Q-ViT** | 4-4 | 44.1 | 2.2 | **83.0** | **96.1** |
| DeiT-B | LSQ | 3-3 | 33.4 | 1.4 | 79.0 | 94.5 |
| | Baseline | 3-3 | 33.4 | 1.4 | 79.3 | 94.9 |
| | **Q-ViT** | 3-3 | 33.4 | 1.4 | **81.0** | **95.1** |
| | LSQ | 2-2 | 22.7 | 0.8 | 70.3 | 88.6 |
| | Baseline | 2-2 | 22.7 | 0.8 | 70.4 | 88.8 |
| | **Q-ViT** | 2-2 | 22.7 | 0.8 | **74.2** | **92.2** |
| | Full-precision | 32-32 | 114.2 | 4.5 | 81.2 | 95.5 |
| | LSQ | 4-4 | 14.6 | 0.6 | 80.2 | 95.2 |
| | Baseline | 4-4 | 14.6 | 0.6 | 80.5 | 95.4 |
| | **Q-ViT** | 4-4 | 14.6 | 0.6 | **82.5** | **97.3** |
| Swin-T | LSQ | 3-3 | 11.2 | 0.3 | 79.7 | 94.9 |
| | Baseline | 3-3 | 11.2 | 0.3 | 79.8 | 95.1 |
| | **Q-ViT** | 3-3 | 11.2 | 0.3 | **80.9** | **96.1** |
| | LSQ | 2-2 | 7.7 | 0.2 | 70.4 | 88.8 |
| | Baseline | 2-2 | 7.7 | 0.2 | 70.6 | 89.0 |
| | **Q-ViT** | 2-2 | 7.7 | 0.2 | **74.7** | **92.5** |
| | Full-precision | 32-32 | 199.8 | 8.7 | 83.2 | 96.2 |
| | LSQ | 4-4 | 7.0 | 1.1 | 82.5 | 97.1 |
| | Baseline | 4-4 | 7.0 | 1.1 | 82.9 | 97.3 |
| | **Q-ViT** | 4-4 | 7.0 | 1.1 | **84.4** | **98.3** |
| Swin-S | LSQ | 3-3 | 5.5 | 0.6 | 80.6 | 95.7 |
| | Baseline | 3-3 | 5.5. | 0.6 | 80.9 | 95.9 |
| | **Q-ViT** | 3-3 | 5.5 | 0.6 | **82.7** | **97.5** |
| | LSQ | 2-2 | 3.9 | 0.3 | 72.4 | 90.2 |
| | Baseline | 2-2 | 3.9 | 0.3 | 72.7 | 90.6 |
| | **Q-ViT** | 2-2 | 3.9. | 0.3 | **76.9** | **94.9** |

# 6   Conclusion

In this paper, we introduce Q-ViT to improve the fully quantized ViTs with high compression ratio and competitive performance. We first build a theoretical framework of fully quantized ViT and analysis the bottlenecks of the fully quantized ViT baseline. Then we introduce Information Rectification Module and Distribution Guided Distillation to Q-ViT for performance improvement. Our proposed Q-ViTs achieve comparable performance with full-precision counterparts with ultra-low bit weights and activations. Our work gives an insightful analysis and effective solution about the crucial issues in ViT full quantization, which blazes a promising path for the extreme compression of ViT.

# 7 Acknowledgement

This work was supported in part by the National Natural Science Foundation of China under Grant 62076016, under Grant 62206272 and 61827901, Beijing Natural Science Foundation-Xiaomi Innovation Joint Fund L223024, Foundation of China Energy Project GJNY-19-90.

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
