# OpenReview forum: "Q-ViT: Accurate and Fully Quantized Low-bit Vision Transformer"
_NeurIPS.cc/2022/Conference — NeurIPS 2022 Accept_

### Official Review · Reviewer_RZGP · 2022-07-08

**Rating:** 7
**Confidence:** 5
**Soundness:** 3 good
**Presentation:** 3 good
**Contribution:** 3 good

**Summary:**

In this paper, Q-ViT is proposed, an effective fully quantized scheme for vision transformer. This paper first introduces the bottleneck for severe performance drop and then present the Information Rectification Module (IRM) and Distribution Guided Distillation (DGD) to bulld a high-performance fully quantized vision transformers. And the experimental results demonstrate the superiority of Q-ViT.

**Questions:**

1. Please prove why maximizing the mutual information between quantized and full-precision representations is equivalent to maximizing the information entropy $H(Q_x)$, there is nothing new in the the disussion of supplementary material.

2. The learnable parameters $\gamma_q$, $\beta_q$ and $\gamma_k$ and $\beta_k$ are updated with the total loss, they could get a better information as Figure 4, but how to prove that they could achieve the state of information maximization. The Information Rectification Module (IRM) seems a loss-aware method instead a strong constrain to keep the max information.



**Limitations:**

Yes

**Strengths And Weaknesses:**

Strengths
1. The analysis of the bottleneck for severe performance drop is clear.
2. The experimetal results are state-of -the-art.
3. The paper is well-written and easy to understand.
4. The code and checkpoints are open sourced.

Weaknesses
1. When the deterministic quantization function is applied to quantized ViT, such objective is equivalent to maximizing the information entropy. The reason is not clearly described.
2. The experiments are sufficient but the proof of these two techniques is a bit weak.

---

> ### Author Response · Authors · 2022-08-01
> **To Reviewer RZGP**
>
> Thanks for your constructive comments and support. The concern is addresssed point to point below.
>
> **Q1**: When the deterministic quantization function is applied to quantized ViT, such objective is equivalent to maximizing the information entropy. The reason is not clearly described.
>
> **A1**: As described in Line 14-18 of [1], *it is shown that, for a class of signal distributions, which includes the Gaussian, the quantizers with maximum output entropy (MOE) and minimum average error (MAE) are approximately the same within a multiplicative constant.* Thus the process of minimizing the error between full-precision values and quantized values is equivalent to maximizing the information entropy of the quantized values. We update the relevant part of our paper and highlight them in blue in the new version.
>
> **Q2**: The experiments are sufficient but the proof of these two techniques is a bit weak.
>
> **A2**: Following [1], given a quantizer with $a$ bits by quantizing full-precision values ${\bf x}$ into a set $\mathcal{Q} = \{[Q_1 = -2^{a-1}, Q_2, \cdots, Q_{N - 1}, Q_N = 2^{a-1}-1]\}, N = 2^a$, where the quantized value $Q({\bf x})$ is in the set $\mathcal{Q}$. The average mutual information $I({\bf x}; Q({\bf x}))$ is
>
> &emsp;&emsp;&emsp;&emsp;&emsp;&emsp;&emsp;&emsp;&emsp;&emsp;&emsp;&emsp;&emsp;&emsp;&emsp;&emsp;&emsp;&emsp;&emsp;&emsp;&emsp;$I({\bf x}; Q({\bf x})) = H(Q({\bf x})) - H(Q({\bf x}) | {\bf x}) = H(Q({\bf x})).$
>
> For fixed bit width $a$, $I({\bf x}; Q({\bf x}))$ is maximized by choosing
>
> &emsp;&emsp;&emsp;&emsp;&emsp;&emsp;&emsp;&emsp;&emsp;&emsp;&emsp;&emsp;&emsp;&emsp;&emsp;&emsp;&emsp;&emsp;&emsp;&emsp;&emsp;&emsp;&emsp;$p_k = Pr\{Q({\bf x} = Q_k)\} = \frac{1}{2^a}, k \in {1, \cdots, N},$
>
> where $Pr\{\cdot\}$ denotes the probability.
> The process of minimizing the average error (MAE) between full-precision values and quantized values is written as
>
> &emsp;&emsp;&emsp;&emsp;&emsp;&emsp;&emsp;&emsp;&emsp;&emsp;&emsp;&emsp;&emsp;&emsp;&emsp;&emsp;&emsp;&emsp;&emsp;&emsp;&emsp;&emsp;&emsp;$E_{\theta} = \sum_{k=1}^N \int_{Q_k}^{Q_{k+1}}p_k \cdot |{\bf x} - Q({\bf X}) |^{\theta} d{\bf x}.$
>
> And as mentioned in [1,2], an approximate relationship for the MAE objective with Gaussian distribution is
>
> &emsp;&emsp;&emsp;&emsp;&emsp;&emsp;&emsp;&emsp;&emsp;&emsp;&emsp;&emsp;&emsp;&emsp;&emsp;&emsp;&emsp;&emsp;&emsp;&emsp;&emsp;&emsp;&emsp;$\int_{Q_k}^{Q_{k+1}} p_k^* d{\bf x} \cong \frac{1}{N},$ &emsp;&emsp;&emsp;&emsp;$p_k^* = A p_k^{\frac{1}{1 + \theta}},$
>
> where $A$ is a constant. Thus the quantization process of minimizing quantization error is approximately the same as maximizing the information entropy. We propose IRM for modifying the information entropy and distributions of the quantized representations in the forward process, and DGD for minimizing the information gap between full-precision representations and quantized counterparts in the backward process. In this case, we prove that our method improves the performance of quantized ViT through maximizing the information entropy of the quantized representations.
> We add these proof according to your comments in the new version of Supplementary file and highlight them in blue.
>
> **Q3**: Please prove why maximizing the mutual information between quantized and full-precision representations is equivalent to maximizing the information entropy $H(Q_x)$, there is nothing new in the the discussion of supplementary material.
>
> **A3**: Following [1], we prove that maximizing the mutual information between quantized and full-precision representations is equivalent to maximizing the information entropy $H(Q_x)$. Please refer to **A2** above.
>
> **Q4**: The learnable parameters $\gamma_q$, $\beta_q$ and $\gamma_k$ and $\beta_k$ are updated with the total loss, they could get a better information as Figure 4, but how to prove that they could achieve the state of information maximization. The Information Rectification Module (IRM) seems a loss-aware method instead a strong constrain to keep the max information.
>
> **A4**: We first change the information entropy of ${\bf q}$ and ${\bf k}$ in quantized ViT into $\frac{1}{2}\log 2 \pi e [\gamma^2_{\bf q}(\sigma^2_{\bf q} + \epsilon_{\bf q})]$ and $\frac{1}{2}\log 2 \pi e [\gamma^2_{\bf k} (\sigma^2_{\bf k} + \epsilon_{\bf k})]$ through IRM, as described in Eq. (8) and Eq. (9) in the paper. Then we distill the $\tilde{\bf q}$ and $\tilde{\bf k}$ for minimizing the information gap between full-precision attention and quantized counterparts through DGD *i.e.,* the gap between $\mathcal{H}(Q_a({\bf x}))$ and $\mathcal{H}({\bf x})$), which maximizes the information entropy of quantized representations to approximation the information entropy of full-precision counterparts.
>
> [1] Messerschmitt. Quantizing for maximum output entropy (corresp.). IEEE Transactions on Information Theory, 17(5):612–612, 1971.
>
> [2] Bernard Smith. Instantaneous companding of quantized signals. Bell System Technical Journal, 36(3):653–709, 1957.

---

> > ### Comment · Reviewer_RZGP · 2022-08-09
> > **Thanks for your response**
> >
> > I have read all the reviews and author response, the authors made significant efforts to address all the raised concerns.  I would keep my decision as accept.

---

> > > ### Author Response · Authors · 2022-08-09
> > > **Thanks for reviewing**
> > >
> > > Thanks again for your valuable time and constructive comments in reviewing our paper. We will further revise and polish our final version towards publication.

---

### Official Review · Reviewer_RZMx · 2022-07-11

**Rating:** 7
**Confidence:** 4
**Soundness:** 4 excellent
**Presentation:** 3 good
**Contribution:** 3 good

**Summary:**

This paper focused on Vision Transformers quantization aware training methods. In this paper, the bottleneck of QAT for ViT are firstly studied which are mainly caused by the information distortion in MHSA. This paper thus propose IRM and DGD scheme to solve such bottleneck which retains the performance of quantized ViT from full-precision counterparts.

**Questions:**

See weakness above.

**Limitations:**

-

**Strengths And Weaknesses:**

Strength:

1. The paper is technically sound and easy to understand.

2. The experiments show that the proposed method is effctive.

Weakness:

1. How is the “6.14x”in Abstract calculated?

2. In L155, how is the distance between each attention weight calculated?

3. In L191, learning rates of learnable gamma and beta are not clarified, are they the same as the learning rate of the whole network?

4. In Eq. 10, why just query and key are distilled, instead of query, key and value?

5. How are alpha_x and alpha_w learned through training? Are the training settings the same as other learnable parameters?

6. In Eq. 7, after maximizing the information entropy, q_x should follow an uniform distribution, however, in the aforementioned figure 4, the query and key may follow a Gaussian distribution. This seems to be conflicting.

7. In Eq. 8, how are the mu and sigma calculated? Are they channel-wise or layer-wise?

8. How is Eq. 9 derived? Especially the part“1/2 log2πe[gamma^2(sigma^2 + epsilon)]”?

---

> ### Author Response · Authors · 2022-08-01
> **To Reviewer RZMx**
>
> Thanks for your feedback. We address your concern point by point below.
>
> **Q1**: How is the “6.14x”in Abstract calculated?
>
> **A1**: We calculated the acceleration rate through FLOPs. Tthe FLOPs of 4-bit Q-DeiT-S is 0.7, while the FLOPs of full-precision model is 4.3 (6.14 $\times$ of 0.7).
>
> **Q2**: In L155, how is the distance between each attention weight calculated?
>
> **A2**: We calculate the distance between each attention weight matrix through l2 distance. We update the description in the new version and highlight  them in blue.
>
> **Q3**: In L191, learning rates of learnable gamma and beta are not clarified, are they the same as the learning rate of the whole network?
>
> **A3**: The learning rates of learnable $\gamma$ and $\beta$ are  same as the whole network. We update the description in the new version and highlight  them in blue.
>
> **Q4**: In Eq. 10, why just query and key are distilled, instead of query, key and value?
>
> **A4**: In Supplementary file, we conduct the ablation experiments of distilling different parts of query, key and value. Thus, we distill query and key only for better performance in the main results.
>
> **Q5**: How are $\alpha_x$ and $\alpha_w$ learned through training? Are the training settings the same as other learnable parameters?
>
> **A5**: $\alpha_x$ and $\alpha_w$ are updated through back propagation and the training settings  are  same as other parameters.
>
> **Q6**: In Eq. 7, after maximizing the information entropy, $q_x$ should follow an uniform distribution, however, in the aforementioned figure 4, the query and key may follow a Gaussian distribution. This seems to be conflicting.
>
> **A6**: As described in Line 15-18 in Supplementary file, $q_{\bf x}$ is the possible values of $Q_a({\bf x})$ (which is $Q_a({\bf q})$ or $Q_a({\bf k})$ in different conditions) with probability $p(\cdot)$. Note that when the information entropy maximizing, all possible quantized values $q_{\bf x} \in \{-Q_n^a, -Q_n^a + 1, \cdots, Q_p^a\}$ intend to follow an uniform distribution, while the reconstructed $\hat{\bf x}$ roughly follow a Gaussian distribution.
>
> **Q7**: In Eq. 8, how are the mu and sigma calculated? Are they channel-wise or layer-wise?
>
> **A7**: In Eq. 8, the mean and the variance of ${\bf q}$ and ${\bf k}$ are calculated based on the whole query or key matrix, *i.e.*, layer-wise.
>
> **Q8**: How is Eq. 9 derived? Especially the part “$\frac{1}{2} \log 2\pi e[\gamma^2(\sigma^2 + \epsilon)]$”?
>
> **A8**: ${\bf q}$ and ${\bf k}$ follow Gaussian distribution, according to the Gaussian hypothesis in \cite{qin2022bibert}. Thus, the information entropy of $\mathcal{H}(Q({\tilde{\bf q}}))$ and $\mathcal{H}(Q({\tilde{\bf k}}))$ is derived based on the information entropy of Gaussian distribution.

---

### Official Review · Reviewer_HQzf · 2022-07-11

**Rating:** 8
**Confidence:** 5
**Soundness:** 3 good
**Presentation:** 4 excellent
**Contribution:** 3 good

**Summary:**

This paper studies the quantization of the Vision Transformer model, especially in the low-bit case. For the problem of large quantization accuracy loss at low bits, two improvements, IRM and DGD, are proposed in this paper, respectively. Concretely, IRM maximizes the information entropy of the quantized representation. DGD supervises the model with other patch-based similarity pattern matrices.

**Questions:**

1.	This paper alleviates the problem of accuracy loss caused by ViT quantization to some extent by two improvement measures. However, I expect that the author dig deeper into the differences between ViT and CNN quantization and the differences between transformer quantization in vision and NLP.
2.	I notice that this work applies Adam optimizer with weight decay set as 0, which is not a regular setting. I suggest the author add extra ablation studies on it to make the paper more complete.


**Limitations:**

The limitations have been well addressed.

**Strengths And Weaknesses:**

Strengths:
1.	The quantization of ViT models is an important research topic. In this paper, the accuracy loss caused by ViT quantization is mitigated to some extent.
2.	The article is well written and well finished. It is easy to follow.

Weaknesses:
1.	The analysis of vit quantification could be explained in depth:
(a) this paper argues that `a direct quantization method leads to the information distortion’ in Line 45. The approach proposed in this paper does not improve this phenomenon either (e.g. 1.2268 in Fig1(b) v.s. 1.3672 in Fig5(b) for Block.3. The variance difference is even larger with the proposed approach).
(b) The quantization of MHSA introduces a large loss of precision, which has been found in transformer quantization in the NLP (such as Q-BERT, Q8BERT, BinaryBERT, FullyBinaryBert, etc.) and is not unique to the ViT model.
2.	Some minor problems:
(a)	In Fig2, the tilde hat of k is too small. It should be inconsistent with q’s hat.
(b)	In Equation 9, $Q_k$ might be $Q(k)$ to be consistent with $Q(q)$.

---

> ### Author Response · Authors · 2022-08-01
> **To Reviewer HQzf**
>
> Thanks for your feedback. We address your concern point by point below.
>
> **Q1(a)**: this paper argues that `a direct quantization method leads to the information distortion’ in Line 45. The approach proposed in this paper does not improve this phenomenon either (e.g. 1.2268 in Fig. 1(b) v.s. 1.3672 in Fig. 5(b) for Block.3. The variance difference is even larger with the proposed approach).
>
> **A1(a)**: Sorry for the confusion,  the variance difference might not mean that the information distortion phenomenon has not been improved. As shown in Fig. 5, the distribution in Q-ViT with IRM follows a distribution closer to Gaussian distribution compared to the vanilla baseline, although the difference of variance is larger. Differently, the distribution of baseline is different from the Gaussian distribution generated from the mean and variance of corresponding values (red line vs. blue shadow in Fig. 1(b)). With the Gaussian distribution hypothesis (supported by [1]), our Q-ViT with IRM mitigate the information distortion phenomenon through re-distributing attention module in the Q-ViT towards Gaussian distribution of the full-precision counterparts.
>
> **Q1(b)**: The quantization of MHSA introduces a large loss of precision, which has been found in transformer quantization in the NLP (such as Q-BERT, Q8BERT, BinaryBERT, FullyBinaryBert, etc.) and is not unique to the ViT model.
>
> **A1(b)**: This phenomenon is not unique to the ViT model, however previous quantization methods in the NLP *e.g.*, Q-BERT, Q8BERT, BinaryBERT and FullyBinaryBert fail to effectively address such bottleneck. Differently, our work improves the quantized ViTs from a new perspective. The IRM and DGD can also be used for BERT-based models on NLP, which will be further explored in our future work.
>
> **Q2**: Some minor problems: (a) In Fig2, the tilde hat of k is too small. It should be inconsistent with q’s hat. (b) In Equation 9, $Q_k$ might be $Q(k)$ to be consistent with $Q(q)$.
>
> **A2**: We correct these typos in the new version  highlight them in blue.
>
> **Q3**: This paper alleviates the problem of accuracy loss caused by ViT quantization to some extent by two improvement measures. However, I expect that the author dig deeper into the differences between ViT and CNN quantization and the differences between transformer quantization in vision and NLP.
>
> **A3**: (1) *Differences between ViT and CNN quantization*:
>
> CNNs calculate features from  local regions   based on convolution, while ViT calculate features based  on global information via inner product. That means that local feature quantization error will only affect part of CNN features, but affect more on the global feature  extraction of  ViT. Also, previous CNN quantization methods  fail to address the severe performance degeneration problem on ViT.
>
> (2) *Differences between ViT and BERT-based NLP models quantization*:
>
> Information density is different between language and vision. Texts are highly semantic and information-dense, while images are natural signals with heavy spatial redundancy. As a result, existing quantization methods in NLP  are  generally different from those in vision.
>
> We add these discussion according to your comments in the new version of Supplementary file and highlight them in blue.
>
> **Q4**: I notice that this work applies Adam optimizer with weight decay set as 0, which is not a regular setting. I suggest the author add extra ablation studies on it to make the paper more complete.
>
> **A4**: We set weight decay as 0 for a better performance. The more detailed ablation experiments are listed in the table below. We update the description in the new version of our Supplementary file and highlight  them in blue.
>
> |&emsp;&emsp;Weight decay&emsp;| Bits  | Top-1 | Bits  | Top-1 | Bits  | Top-1 |
> |:-:|:-:|:-:|:-:|:-:|:-:|:-:|
> |0|4-4|**80.9**|3-3|**79.0**|2-2|**72.0**|
> |1e-4|4-4|80.4|3-3|78.6|2-2|71.5|
> |2.5e-5|4-4|80.6|3-3|78.7|2-2|71.6|
> |1e-5|4-4|80.6|3-3|78.7|2-2|71.8|
> |1e-6|4-4|**80.9**|3-3|78.9|2-2|71.8|
>
> [1] Haotong Qin, Yifu Ding, Mingyuan Zhang, Qinghua Yan, Aishan Liu, Qingqing Dang, Ziwei Liu, and Xianglong Liu. Bibert: Accurate fully binarized bert. In Proc. of ICLR, pages 1–24, 2022.

---

### Official Review · Reviewer_BbJC · 2022-07-11

**Rating:** 6
**Confidence:** 4
**Soundness:** 3 good
**Presentation:** 3 good
**Contribution:** 3 good

**Summary:**

This paper proposed a novel and efficient quantization method for Vision Transformers. The authors first identify the bottleneck low-bit quantized Vision Transformers which comes from the information distortion of the low-bit quantized self-attention map. The authors then develop an IRM and a DGD scheme for fully quantized Vision Transformers based on the aforementioned bottleneck, which leads to a fully quantized Vision Transformers.

**Questions:**

Please see above.

**Limitations:**

Please see above.

**Strengths And Weaknesses:**

1.In Line 61, what is the actual meaning of “quantified representations”?

2.In section 4.1 “Architecture bottleneck”, how are these experiments trained? the same as the main experiments settings?

3.In Line 155, “distance”should be “distances”.

4.Font sizes of the legend in Figure 4 are inconsistent and should be larger.

5.Are the IRM applied before quantizing attention module, or after quantizing? Such two situations may lead to different results.

6.In section 5.1, authors mentioned that the Q-ViT is initialized from pertained full-precision counterparts. However, how are the alpha and zero point initialized?

7.In Eq. 7, what does “q_x is the random quantized variables in Q_a(x)” mean? What are the possible values of “q_x”?

8.In Eq. 8, after introducing IRM, the module calculate the mean and variance of Q and K in each block for each forward process, will this affect the speed of inference?

---

> ### Author Response · Authors · 2022-08-01
> **To Reviewer BbJC**
>
> Thanks for your constructive and supportive comments.
>
> **Q1**: In Line 61, what is the actual meaning of “quantified representations”?
>
> **A1**: "quantized representations" in Line 61 represents the quantized query and key *i.e.*, $Q_a({\bf q})$ and $Q_a({\bf k})$ in our MHSA module of each layer in a ViT model.
>
> **Q2**:  In section 4.1 “Architecture bottleneck”, how are these experiments trained? the same as the main experiments settings?
>
> **A2**: These ablation experiments are conducted the same as experiments in Tab. 2. We added this description in the new version and highlight them in blue.
>
> **Q3**:  In Line 155, “distance”should be “distances”.
>
> **A3**:  We revise  the typo and highlight in blue.
>
> **Q4**:  Font sizes of the legend in Figure 4 are inconsistent and should be larger.
>
> **A4**:  We update the figure in the new version.
>
> **Q5**:  Are the IRM applied before quantizing attention module, or after quantizing? Such two situations may lead to different results.
>
> **A5**: The IRM is applied before quantizing the query and key in each MHSA module, which modifying the information entropy and distributions of the query and key for minimizing the information gap between full-precision representations and quantized counterparts. If IRM is applied after quantizing the query and key, the information distortion already exists and cause the essential global information to be distorted by the quantization operation. Thus, we appliy IRM before quantizing the attention module.
>
> **Q6**: In section 5.1, authors mentioned that the Q-ViT is initialized from pertained full-precision counterparts. However, how are the alpha and zero point initialized?
>
> **A6**: $\alpha$ and $z$ are initialized as random parameters in our implementation.
>
> **Q7**:: In Eq. 7, what does “$q_x$ is the random quantized variables in $Q_a(x)$” mean? What are the possible values of “$q_x$”?
>
> **A7**: As described in our Supplementary file, $q_{\bf x}$ is the possible values of $Q_a({\bf x})$ (which is $Q_a({\bf q})$ or $Q_a({\bf k})$ in different conditions) with probability $p(\cdot)$. For example, if quantizing ${\bf x}$ into $a$ bits, the possible discrete values of $q_x$ are $[-2^a, -2^a + 1, \cdots, 2^{a-1} - 1, 2^{a-1}]$.
>
> **Q8**: In Eq. 8, after introducing IRM, the module calculate the mean and variance of Q and K in each block for each forward process, will this affect the speed of inference?
>
> **A8**: The IRM operation will not be involved into the inference process, thus it will not affect the speed of inference.

---

### Author Response · Authors · 2022-08-02
**To ACs and Reviewers**

We would like to sincerely thank the ACs and all reviewers for the positive comments and valuable suggestions. We have carefully addressed all comment point to point in the following. We have rearranged our manuscript and supplementary material according to the comments Newly added or modified texts are highlighted in blue in the revised manuscript. We wish our revision can satisfy the requirements of all Reviewers.

---

### Meta-Review · Area_Chair_xRdN · 2022-08-26

**Recommendation:** Accept
**Confidence:** Certain

**Metareview:**

This paper proposes a novel method for Vision Transformers quantization. The IRM and DGD scheme is developed to solve the bottleneck of low-bit quantized Vision Transformers. All the reviewers agree that the proposed method is novel and effective. The concerns and questions are well addressed during the rebuttal period. The overall quality is clearly above the bar, and thus the paper should be accepted for publication.

**Award:**

No

---

### Decision · Program_Chairs · 2022-09-14

Accept